# N-Bref: A High-fidelity Decompiler Exploiting Programming Structures

## Abstract

Binary decompilation is a powerful technique for analyzing and understanding software, when source code is unavailable. It is a critical problem in the computer security domain. With the success of neural machine translation (NMT), recent efforts on neural-based decompiler show promising results compared to traditional approaches. However, several key challenges remain: (i) Prior neural-based decompilers focus on simplified programs without considering sophisticated yet widely-used data types such as pointers; furthermore, many high-level expressions map to the same low-level code (expression collision), which incurs critical decompiling performance degradation; (ii) State-of-the-art NMT models (e.g., transformer and its variants) mainly deal with sequential data; this is inefficient for decompilation, where the input and output data are highly structured. In this paper, we propose *N-Bref*[1], a new framework for neural decompilers that addresses the two aforementioned challenges with two key design principles: (i) N-Bref designs a structural transformer with three key design components for better comprehension of structural data – an assembly encoder, an abstract syntax tree encoder, and a tree decoder, extending transformer models in the context of decompilation. (ii) N-Bref introduces a program generation tool that can control the complexity of code generation and removes expression collisions. Extensive experiments demonstrate that N-Bref outperforms previous neural-based decompilers by a margin of 6.1%/8.8% accuracy in datatype recovery and source code generation. In particular, N-Bref decompiled human-written Leetcode programs with complex library calls and data types in high accuracy.

## 1 Introduction

Decompilation, which is a process of recovering source code from binary, is useful in many situations where it is necessary to analyze or understand software for which source code is not available. For example, decompilation is highly valuable in many security and forensics applications (Lin et al. (2010); Lee et al. (2011); Brumley et al. (2011)). Given a binary executable, an ideal decompiler generates the high-level program that preserves both the semantics and the functionality of the source code. However, this process is difficult as the data structure and semantics are largely destroyed or obfuscated during the compilation. Inspired by remarkable performance in neural machine translation (NMT) tasks (Liu et al. (2019); Vaswani et al. (2017); Dai et al. (2019); Devlin et al. (2018); Dong & Lapata (2016)), recent works (Fu et al. (2019); Katz et al. (2019)) leverage NMT model for neural-based decompilation and achieve promising performance on small code snippets.

To make neural-based decompilation useful in practice, many challenges remain: **(C1)** Current state-of-the-art neural architectures for machine translation – transformer (Vaswani et al. (2017)) or its variants (Dai et al. (2019); Devlin et al. (2018); Liu et al. (2019)) – focused on sequential data (e.g., language), while neural decompilers deal with data with intrinsic structures (e.g., tree/graph) and long-range dependencies. **(C2)** The main decompilation task consists of many sub-tasks (e.g., datatype recovery, control/dataflow recovery). Training one neural network cannot solve them all. **(C3)** Practical data types (e.g., pointers) are not modeled and compiling configurations need to be known beforehand (Fu et al. (2019)). **(C4)** Due to a lack of unification in terms of library usage, variable type, and/or control-flow complexity, a simple crawling from public repositories does not

---

[1] N-Bref is the abbreviation for "neural-based binary reverse engineering framework"

work well. Source code of different styles can be compiled into identical binary code (i.e., "expression collision" or EC) and yield issues when evaluating decomplied code against original source code. To our best knowledge, no code generation toolkit with configurable code complexity exists.

In this paper, we present *N-Bref*, an end-to-end neural-based decompiler framework that learns to decompile the source code to assembly. For **(C1)**, we design a *back-bone structural transformer* by incorporating inductive Graph Neural Networks (GNNs) (Hamilton et al. (2017)) to represent the low-level code (LLC) as control/dataflow dependency graphs and source code as Abstract Syntax Tree (AST). To better model long-range correlations in the structural representations, we add a graph neural network after each of the self-attention layers in the transformer. The AST decoder expands the AST of the source code in a tree fashion to better capture the dependency of each predicted node. Also, we adopt memory augmentation (Cornia et al. (2019)) and new tokenizing methods to improve the scalability of our neural networks with the growing size of programs. The backbone network is learned to iteratively generate AST for source code from structured representation of assembly.

For **(C2)** and **(C3)**, we decouple decompilation into two sub-tasks: *data type solver* (**DT-Solver**) and *source code generator* (**SC-Gen**), both use the same backbone structural transformer with different parameters. The output of the data type solver is used as the decoder input of the source code generation. For **(C4)**, we design a dataset generator to generate training data, test and analyze the performance of different design principles across configurable code complexity. Different from conventional dataset generators (Yang et al. (2011); IntelC++compiler (2017)) used in programming language studies, our generator produces similar code styles as those written by human programmers, has unified source code representation that avoids EC, has configurable complexity and data types to facilitate factor analysis, and is specifically designed for learning-based methodologies.

Extensive experiments show that on our new metrics, N-Bref outperforms transformer baseline/previous neural-based decompiler (Fu et al. (2019)) by 3.5%/6.1% and 5.5%/8.8% in data type recovery and source code generation tasks, respectively. Furthermore, on 5 human-written Leetcode solutions, N-Bref shows 4.1%/6.0% and 6.0%/9.7% margins over transformer/previous neural decompiler in data type recovery and source code generation, respectively. We also perform a comprehensive study of the design component in neural-based decompiler across different dataset configurations. In summary, this paper makes the following contributions:

• We construct an end-to-end decompilation system by integrating a LLC Encoder, an AST encoder, an AST decoder, and a set of novel embedding methods in a holistic manner. Our new architectures bridge the gap between low-level code and high-level code by transforming both of them into a graph space.

• We perform a comprehensive analysis of the influence of each neural-based decompiler design component to the overall program recovery accuracy across different dataset configurations. We corroborate the design performance on various generated benchmarks and Leetcode tasks.

• We boost decompilation performance by decomposing the decompilation process into separate tasks, data type recovery and AST generation. In addition, we present corresponding new metrics to evaluate data type recovery and source code generation.

• We develop the first dataset generation tool for neural-based decompiler development and testing. It randomly generates programs with configurable complexity and data types; it also unifies source code representation to prevent "expression collision".

## 2    PRELIMINARIES OF DECOMPILERS

Decompilation takes an executable file as input and attempts to create high-level source code that are more semantically meaningful and can be compiled back. Figure 1 shows a low-level code snippet disassembled from a stripped binary and the corresponding high-level program.

A commonly used low-level code (LLC) is assembly (ASM). An assembly program is a sequence of instructions that can be executed on a particular processor architecture (e.g. MIPS, x86-64). The first token for each instruction is called an "opcode", which specifies the operation to be performed by the instruction. Many instructions in a program operate on processor registers (a small amount of fast storage in the processor) or instant values to perform arithmetic operations, such as shifting (e.g.*shl*, *shr*), floating-point multiplications (e.g. *mulss*), etc. Other instructions include (1)

---

[2]  Complete assembly code and graph are shown in Appendix H & I.

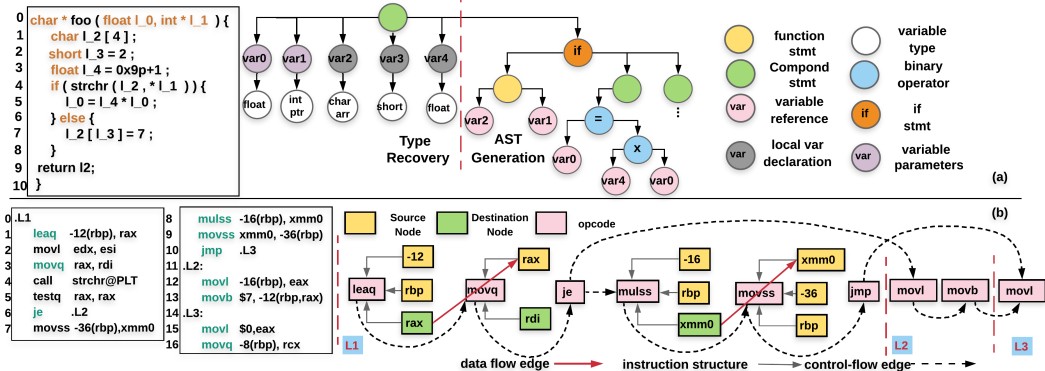

**Figure 1:** An example of (a) source code to AST conversion. Note that the pseudo 'statement' or 'stmt' nodes are added by the compiler during AST conversion. (b) assembly (x86-64) to graph[2].

memory instructions that load (Figure 1(b) Line 1) or store (Line 9) data from memory/register to register/memory; (2) branch instructions that conditionally (Line 6) or unconditionally (Line 10) redirect program execution to a different sequence.

Each instruction has a certain internal structure, depending on the opcode. For example, in Line 8 of Figure 1(b), the first operand is a floating-point value in the memory and $multss$ multiplies the value with the destination register ($xmm0$) and stores the value back to $xmm0$. Besides, connections also exist between instructions: (i) branch instructions (e.g., $je$, $jmp$) reveal the 'control flow' of the high-level program; (ii) the register which stores the new value of $multss$ (Line 8) is consumed later as a source register (Line 9). These data movements reveal the 'data flow' of the program. In this paper, we formulate the low-level instructions as a graph using the instruction structure, control-flow and data-flow between each nodes as shown in Figure 1(b).

High-level programming languages can be represented in its equivalent abstract syntax tree (AST) (Baxter et al. (1998)) during code generation (Figure 1(a)). This representation has many advantages over its sequential representations: (i) adjacent nodes are logically closer in AST compared with sequential representations, (ii) error propagation in sequential expansion can be alleviated in a tree decoder, and (iii) AST grammar helps prevent error predictions.

## 3 N-BREF OVERVIEW

In this section, we provide an overview of our design components with an illustrative example. Figure 2 shows an example of the prediction procedures.

■ **The Backbone Structural Transformer.** Our structural transformer has three components: (1) LLC encoder, (2) AST encoder, and (3) AST decoder (Detailed in Sec. 4). The LLC encoder takes the low-level instructions converted from binaries using disassembler as input. AST encoder takes the input of a previous (partial) AST, and the predictions of AST decoder are AST nodes, which can be equivalently converted to the high-level program. As mentioned earlier, we formulate input low-level code into graphs and high-level code into tree structures.

As the AST of the data declaration is very distinct from the rest of the code (Figure 1(a)) in high-level program, we decompose decompilation into two tasks: data type solver (`DT-Solver`) and source code generator (`SC-Gen`). Both have the backbone structural transformer.

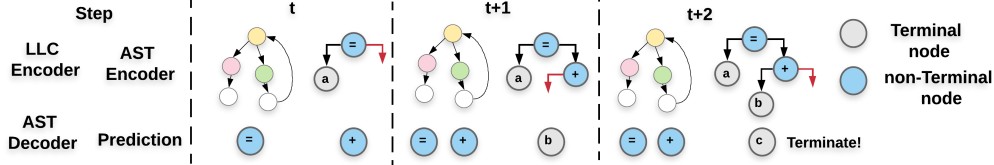

**Figure 2:** An example of prediction procedures in N-Bref pipeline. AST is expanded in a breadth-first manner.

■ **Prediction Procedure.** Figure 1 shows an example of the code recovery process of N-Bref. The assembly graph and high-level AST is the input of LLC encoder and AST encoder. The input of the AST decoder is the tree path from the root node to the expansion node. Initially, a single root-node is fed into the AST encoder/decoder. Once a new node is generated from decoder in each step, we

**Table 1:** Hyper-parameters in data generation

| Hyper-parameters | Description |
|---|---|
| Block depth ($b_{depth}$) | maximal level of nested control dependencies ('loop' or 'branches'). |
| Block size ($b_{size}$) | maximal number of children for a component statement (See Figure 1(a)) for each code block. |
| Block number ($b_{num}$) | maximal number of times to generate the control blocks with depth of $b_{depth}$ |
| Expression complexity ($E_c$) | maximal depth of each line of code |

update the AST and use it as the AST encoder input in the next prediction step. We expand the AST in a breadth-first (BFS) fashion.

AST contains explicit terminal nodes, which are tokens with no child, such as registers, numerics, variable references and variable types. Non-terminal nodes (e.g. binary operator '=') must have children, otherwise there is a syntax error. The branch stop expansion when its leaf nodes are all terminal nodes. Note that during training, we apply 'teacher forcing' by attaching the correct node label into the AST encoder at each step. (See Appendix E for formal algorithm)

■ **Cascading DT-Solver and SC-Gen.** As shown in Figure 1, we divide the AST into two parts: (i) AST of data type and (ii) AST of main code body. Each part is generated using `DT-Solver` and `SC-Gen` respectively. This method allows the network to focus on each task individually and resolves more complicated data types. During testing, `DT-Solver` first generates the left part of the AST in Figure 1, then the `SC-Gen` will continue the expansion from this intermediate results. During training, the initial data type input to the SC-Gen is the program golden.

## 4 METHODOLOGY: DATA GENERATOR

In this section, we detail the data generator designed in N-Bref. We design data generator so that it has no expression collision (EC). For example, 'unary' operators are converted to 'binary' operators (i++ and i=i+1) and all the 'while' loops are transferred into 'for' loops. Experimentally, we observe that our data generator is free of EC and performance improves. EC hurts the performance because (1) the same input assembly can be mapped to multiple equivalent outputs; (2) extra high-level semantics result in extra token dimensions; (3) the training under EC is both difficult and slow due to label adjustment at runtime.

The generator is configurable with multiple hyper-parameters (Table 1), which makes it easy for N-Bref to accommodate with different decompilation tasks. It also allows us to analyze which components in the pipeline improve the scalability and performance (See Sec. 6).

For each data point in the dataset, we sample $b_{depth}^s$, $b_{size}^s$ and $b_{num}^s$ with a uniform distribution between 1 and a user-specific maximal value (Table 1). The number of sampled variables ($var_{num}^s$) of a program is related to $b_{num}^s$ and $b_{depth}^s$ following a Poisson Distribution (Equations in Appendix D). The generator also takes the libraries $lib_{in}$ that are pre-defined by the user as potential generated components. If a function call is sampled, the data generator will filter out the variables that do not match with its input / output types (line 4 Figure 1(a)).

In summary, $b_{depth}$ and $E_c$ control the difficulty of control/data flow, while $b_{size}$ and $b_{num}$ control the length of the code. For example, the code snippet in Figure 1(a) has a configuration of $[E_c, b_{depth}, b_{size}, b_{num}] = [2, 1, 1, 1]$. Note that in N-Bref, the program is compiled using `gcc` with no optimizations. Previous works (Brumley et al. (2013); Lee et al. (2011); Lin et al. (2010)) also disabled optimizations, because many optimizations change variable types and rewrite source code (e.g. loop-invariant optimization) which will result in unfair accuracy evaluations.

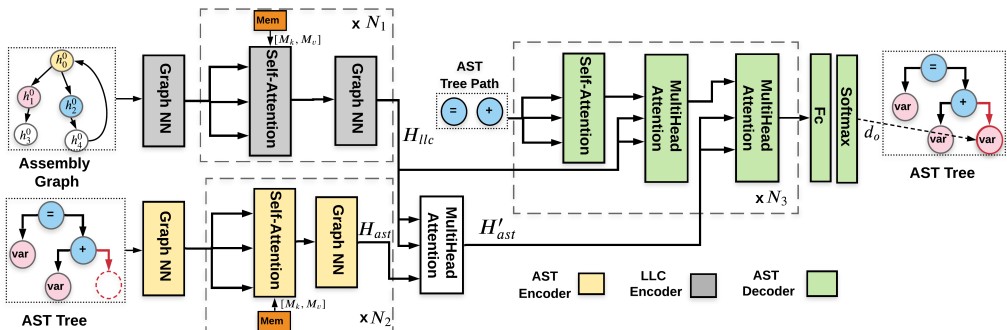

**Figure 3:** The backbone neural architecture design for `DT-Solver` and `SC-Gen`. $N_1, N_2, N_3$ indicates the number of times to repeat the block in the architectures.

## 5 Methodology: Pipeline

Here, we present the details of the backbone structural transformer in N-Bref.

■ **LLC Encoder.** As shown in Figure 1(b), we first formulate the assembly code as graphs by adding the following edges between nodes: (i) between the 'opcode' node of branch instructions and all their possible next instructions (control flow). (ii) between instructions 'opcode' and its 'operands' (i.e., registers / instant values). (iii) between the same register node of different instructions (register read-after-write) which indicates the data-dependency. Different from (Shi et al. (2019)), we do not add redundant pseudo nodes to indicates node positions, because this method is not scalable for long programs due to the exponential size of input using pseudo nodes. Instead, we directly concatenate all one-hot meta-features of a node together, namely the register / instruction type ($var_t$ / $ins_t$), position in the instruction ($n_{pos}$), node id ($n_{id}$) and numerical field ($n_{num}$) during tokenizing process. If the token is a number, we represent it in a binary format to help the transformer generalize to unseen numerical values (e.g., 12 is represented as $n_{num}$=**[0,0,0,0,1,1,0,0]**, a 16-by-1 vector). This representation method can greatly reduce the length of the transformer input and make the decompiler more robust for long programs.

The tokenized vector for each node ($h^0 = [var_t; ins_t; n_{pos}; n_{block}; n_{id}; n_{num}; n_{zeros}]^T$, $h^0 \in \mathbb{R}^{d \times 1}$) are fed into an embedding graph neural network (GNN) – GraphSAGE (Hamilton et al. (2017)), an inductive framework that can generalize representations for unseen graphs. We pad $n_{zeros}$ to the input vector $h^0$ to match the GNN output feature size $d$. Note that to better represent the data flow in assembly code, we leverage a character embedding for registers ($var_t = [c^1; c^2; c^3]$). For instance, if the register is $rax, we would break it into 'r', 'a', 'x'. That is because the naming policy of x86-64 (also MIPS) hardware registers indicates their underlying connections – $eax is the first 32-bit of register $rax and $ax/$ah is the first 16-/8-bit of register $eax.

After getting the assembly graph $V$ and each node's representation $h^0$, each node $v$ (where $v \in V$) aggregates the feature representations of its sampled neighbors:

$$h^l_{N(v)} = max(\sigma(W^l h^l_u + b^l)) \ , \forall u \in N(v) \tag{1}$$

Here, the $h^l_u$ represents the hidden state of a node's neighbours and $N(v)$ represents the set of neighbours of the node $v$. $W^l$ is a trainable matrix ($d$-by-$d$) and $b^l$ is a bias vector, $\sigma$ represents the sigmoid activation function. We choose to use an element-wise max-pooling ($max$) as an aggregator to collect the states from neighbours. The aggregation vector is concatenated with the current state of the node $h^l_u$ as the input to a fully-connected layer to get the new state:

$$h^{l+1}_v = W^{l+1}([h^l_v, h^l_{N(v)}]) \tag{2}$$

Here, $W^{l+1} \in \mathbb{R}^{d \times 2d}$ is the trainable embedding matrix and $h^{l+1}_v$ (a $d$-by-1 vector) is the output of the current layer of GNN. $[\cdot, \cdot]$ is the concatenation operation.

■ **AST Encoder.** The AST encoder encodes the AST tree from the AST decoder to guide future tree expansions. In this work, we treat the AST tree as a graph ($V$) and embed it using GNN in the same way as LLC encoder following Eq. (2)(1). The input of the GNN includes meta-features of the tokenized AST node feature ($n_{feat}$) and a boolean indicating *whether the node is expanded in this step* ($n_{expand}$). The input vector $h_v = [n_{expand}; n_{feat}]$ ($v \in V$) is fed into a GNN and the output ($h'_v$) is added with the positional encoding result:

$$h''_v = h'_v + W_1 h^{depth}_v + W_2 h^{idx}_v \tag{3}$$

Here $h^{depth}_v$ and $h^{idx}_v$ are the one-hot vector of the node's ($v$) depth in the tree and node's position among the parent's children. $W_1$ and $W_2$ are trainable matrices for embedding. The output hidden states is fed into our designed self-attention module (Sec. 5.1). At the end of the AST encoder, we integrate the AST encoder output ($H_{ast}$) and LLC encoder output ($H_{llc}$) using an additional multi-head attention layer with $H_{llc}$ as input $K$ and $V$, and $H_{ast}$ as $Q$ (Figure 3). The result ($H'_{ast}$) will be used for networks downstream.

■ **AST Decoder.** The AST decoder takes the encoding result from the previous stage as input. The querying node of the AST decoder is represented as a path from the root to itself using the same methods proposed in (Zhu et al. (2019); Chen et al. (2018a)). This method reduces the length of the input into the decoder. The results from the low-level code encoder $H_{llc}$ and the AST encoder $H'_{ast}$ are integrated into the decoder using two attention layers following Eq. (4) as shown in Figure 3. The output of the AST decoder is mapped into its output space with dimension $d_o \times 1$ using another fully-connected layer. $d_o$ is the number of possible tokens of high-level code. After the new prediction is generated, we update the AST tree for the next time step (Figure. 2).

## 5.1 Memory and Graph Augmented Self-Attention Module

Decompilation is hardly a word-by-word translation like natural language. Each individual token in low-level code or AST tree must be interpreted in the context of its surrounding semantics. To capture the prior knowledge on programming structures and emphasize the node connections after embedding, we leverage two additional modules (i) memory-augmentation (ii) graph-augmentation in transformer attention. The formal descriptions are shown below.

■ **Memory augmentation.** We propose a memory-augmented attention layer similar to the method in (Cornia et al. (2019)). The input prior information is trained and does not depend on the input data. Traditional transformer's building block is self-attention layer which takes three sets of vectors (queries $Q$, keys $K$ and values $V$) as input. The layer first computes the similarity distribution between $Q$ and $K$ and use the resulted probability to do a weighted sum on $V$. (equations in Vaswani et al. (2017)) In N-Bref, we add two trainable matrices for each head as an extra input to the transformer for memory augmentation. And the computation is adjusted to:

$$H = MultiHead(Q, K, V) = Concat(head_1, ..., head_t)\dot{W}^O \tag{4}$$

$$head_i = Attention(Q', K', V') = softmax(\frac{Q'K'^T}{\sqrt{d}})V' \tag{5}$$

$$\text{where } Q = QW_{qi}, V' = [VW_{vi}, M_{vi}], K = [KW_{ki}, M_{ki}] \tag{6}$$

Here, $t$ is the number of parallel attention heads. ($W_{qi}, W_{ki}, W_{vi}$) are trainable matrices with a dimension of $d \times \frac{d}{t}$. $W^O$ has the dimension of $d \times d$. $M_{vi}$, $M_{ki} \in \mathbb{R}^{d_m \times d}$ and $d_m$ controls the number of slots of the memory. Note that we remove the positional embedding in the original transformer for LLC encoder as the position information is integrated into GNN embedding.

■ **Graph augmentation.** We propose to emphasize the connections of assembly nodes after attention layer. The output of the multi-head attention layer $s$ can be viewed as a matrix $H_t = [h^{s,0}, h^{s,1}, ..., h^{s,N}], H \in \mathbb{R}^{d \times N}$, where $N$ is the number of nodes in the assembly graph. We first convert $H_t$ back to a graph structure using the edge information of assembly code. The new graph with connected hidden states is the input to an additional GNN layer using Eq. (1)(2) after each self-attention layer.

## 6 Evaluation

### 6.1 Experimental Setup

We assess the performance of N-Bref on various benchmarks generated by our dataset generator with various difficulty levels and Leetcode solutions (Problems (2017); details in Appendix B) as shown in Table 2. This binary is disassembled using the GNU binary utilities to obtain the assembly code. To generate the AST for C code, we implement the parser using `clang` compiler for python. Our dataset generator is built on `csmith` (Yang et al. (2011)), which is a code generation tool for compiler debugging. The original tool is not suitable in our cases and thus N-Bref modifies most of the implementation for decompilation tasks. For the neural network setting discussed in Figure 3, we choose $[N_1, N_2, N_3, t] = [3, 2, 2, 8]$ (t is the number of heads used in N-Bref), an embedding dimensionality $d = 256$, and memory slots $d_m = 128$. The training/evaluation is implemented using Pytorch 1.4 and DGL library (Wang et al. (2019)). Details about the training hyper-parameters and settings are included in Appendix A.

■ **Complexity arguments and benchmarks.** This section describes the tasks in our evaluation. We randomly generate 25,000 pairs of high-level programs and the corresponding assembly code in each task for network training (60%), validation (20%) and evaluation (20%). We mainly focus on tuning $b_{depth}$ and $b_{num}$ (see Table. 1). We set $E_c = 3$, $b_{size} = 3$, $bias = 2$ and test $lib_{in}$ with different complexities: **(i)<math.h>** and **(ii)<string.h>**. Function recursion is allowed for code generation. Other than function calls, normal expressions ( "$+, -, *, \backslash, \|, \gg, \&, ==, \wedge$" etc.) are also possible operators during code generation. (Code examples in Appendix C.)

■ **Metrics.** We evaluate the performance of N-Bref using *token accuracy*. For evaluation of `SC-Gen`, we expand the decompiled AST from the root to match the structure of the golden AST ($AST_G$). The *token accuracy* is calculated as:

$$acc = \frac{num(AST = AST_G)}{num(AST_G)} \tag{7}$$

We also show the evaluation result using graph edit distance (Sanfeliu & Fu (1983)) without enforcing the match between the decompiled $AST$ and $AST_G$ on the graph generation in Appendix G.

The metric is fair to evaluate decompilation tasks as we remove all the EC. Eq. 7 is able to evaluate sequence output by treating the sequence as a tree.

For **DT-Solver**, we have two metrics: (i) macro accuracy ($Acc_{mac}$) and (ii) micro accuracy ($Acc_{mic}$). The $Acc_{mac}$ treats **unsigned** and **signed** variables as the same. This is because **unsigned** and **signed** variables have *no difference* in assembly code except through hints from type-revealing functions (for example, the return value of **strlen()** must be an **unsigned integer**). Note that we do not recover numerical values that are directly assigned to a variable and we replace them with a token '$num$'. These values exist in the stack memory or assembly instructions which can be resolved after the AST is correctly recovered using additional analysis methods. One of our future works can leverage the pointer network (See et al., 2017) to directly copy numerical values to the output.

## 6.2 RESULTS

**Performance impact of each design component.** To study the effectiveness of potential design principles, we perform a thorough sensitivity study as shown in Figure 4.

In **SC-Gen**, we observe that with the growth of code length and complexity, each component in N-Bref preserves more performance compared to the baseline transformer. The transformer with LLC encoder shows the least performance degradation when complexity and length increase (as shown by the slope). This is because the GNN in N-Bref connects instructions that are distant from each other to alleviate the performance drop. By expanding the source code into a tree structure (AST encoder+decoder), the result shows that it also prevents the accuracy of degradation.

For **DT-Solver**, increasing $b_{size}$ improves the performance, because short programs do not have enough semantics to identify variable types. Also, the performance declines when increasing the program complexity ($b_{depth}$), and we assume that is because wiring control-flow complicates the analysis of data-flow. Traditional decompiler REWARD shows a large performance drop along the axis of $b_{depth}$. That is because the dynamic analysis in REWARD is for a single control path. As such, it has limited performance among complex control-flow. We also tested many other design options but they cannot achieve better scalability and performance empirically compared to N-Bref.

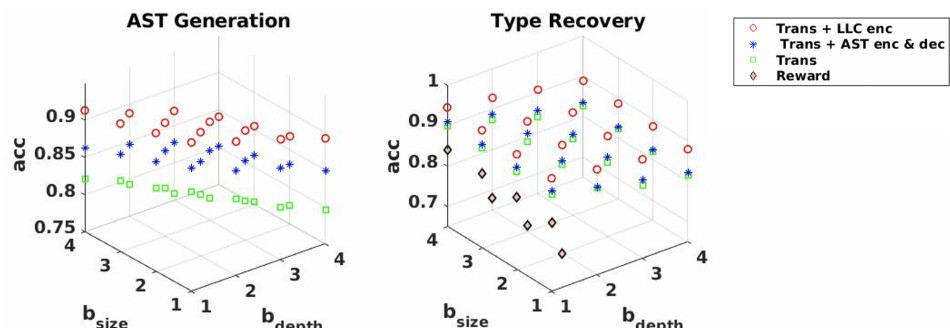

**Figure 4:** Sensitivity analysis of each design component over dataset complexity for <math.h>. Each sample is trained for 10 epochs for simplicity. 'Trans' Refers to the baseline transformer.

**Comparison to previous works.** N-Bref yields the highest token accuracy across all benchmarks (91.1% on average) as shown in Table 2 in both data type recovery and AST generation. N-Bref engenders 5.5% and 8.8% margin over transformer-baseline and Ins2AST, which is a previous neural-based program decompiler (Fu et al. (2019)). The encoder in Ins2AST leverages N-ary Tree LSTM, which cannot collect information for instructions far apart but logically adjacent. Our LLC encoder, on the other hand, leverages GNN embedding and graph augmentation to emphasize node connections. We do not show the results of traditional decompilers (RetDec (2017); Hex-Rays (2017)) as they do not preserve any semantics and achieves very low token accuracy using Eq. 7. (Examples of traditional decompilation results are shown in Fu et al. (2019)).

For type recovery, N-Bref also achieves 3.55% / 6.1% / 30.3% average margin over transformer, Ins2AST, and REWARD respectively. Traditional decompiler REWARD leverages type-revealing instructions and does not consider other low-level representations. Also, REWARD focuses on a single path in the program executed using dynamic analysis. As such, they cannot handle control flow properly. N-Bref uses static analysis and considers all paths during execution.

For baseline transformer and N-Bref, we also present the $Acc_{mic}$ in parentheses of Table 2. The gap between $Acc_{mac}$ and $Acc_{mic}$ is reduced when the program gets longer (reduced from 28.5% to 20.2% for **<math.h>** using N-Bref). That is because longer program have more type-revealing instructions/functions which can help the network to identify data types ('unsigned/signed') correctly. Also, N-Bref shows higher tolerance to the complexity and length growth compared to other works. For $lib_{in}$ =**<string.h>**, the token accuracy drops by 6.1% from the easiest to the hardest settings compared to 9.5% / 8.2% accuracy drop for baseline transformer and Ins2AST, respectively. That is because GNN can gather information from adjacent nodes for large assembly graphs. Also, AST decoders can effectively prevent error propagation through tree expansion when the source code grow larger, unlike sequence generation where early prediction errors affect the later nodes.

We also select 5 Leetcode Problems (2017) solutions in C as an auxiliary test dataset and train a new model with the complexity of ($b_{depth} = 2$, $b_{size} = 4$) using <**string.h**> library. The result shows N-Bref is able to recover real benchmarks and achieves 6% / 9.7% margin over transformer and previous neural-based decompiler. This means N-Bref is able to generalize to human-writing code. The complexity of datasets we generate can cover real-world applications.

**Table 2:** Accuracy (%) comparison between N-Bref and alternative methods in (a) type recovery, and (b) AST generation. Ins2AST is a previous neural-based program decompiler (only code sketch generation stage). REWARD (Lin et al. (2010)) is a traditional framework for type recovery. Lang2logic (Dong & Lapata (2016)) is a sequence-to-tree translator. The baseline is the transformer.

| $(lib, b_{size}, b_{depth})$ | (a) Data Type Recovery $Acc_{mac}(Acc_{mic})$ | | | | | (b) AST Generation (token accuracy) | | | |
|---|---|---|---|---|---|---|---|---|---|
| | REWARD | baseline | lang2logic | Ins2AST+att | N-Bref | baseline | lang2logic | Ins2AST+att | N-Bref |
| (math,1,1) | 85.1 | 96.6 (70.2) | 92.1 | 94.1 | **99.6** (71.1) | 90.3 | 84.3 | 88.6 | **94.5** |
| (math,2,2) | 66.2 | 94.1 (71.4) | 88.6 | 91.3 | **97.3** (72.5) | 87.3 | 81.5 | 85.6 | **92.2** |
| (math,3,3) | 53.1 | 91.9 (73.4) | 86.4 | 88.3 | **95.3** (75.1) | 84.0 | 77.8 | 82.4 | **89.5** |
| (str,1,1) | 82.4 | 95.3 (71.6) | 91.1 | 92.9 | **98.3** (73.3) | 88.6 | 80.8 | 85.1 | **92.9** |
| (str,2,2) | 63.1 | 93.1 (72.5) | 88.3 | 90.8 | **97.0** (74.9) | 84.3 | 75.6 | 81.5 | **90.6** |
| (str,3,3) | 50.9 | 91.5 (73.6) | 84.8 | 88.6 | **95.3** (75.4) | 79.1 | 70.3 | 76.9 | **86.8** |
| leetcode | 73.3 | 91.9 (73.8) | 85.4 | 89.1 | **96.0** (75.9) | 82.3 | 73.1 | 78.6 | **88.3** |

**Table 3:** Ablation study of N-Bref on AST generation. '-ensemble' refers to disable the separation of data type recovery and AST generation.

| $(lib, b_{size}, b_{depth})$ | -GNN after attention | -node representation | -positional encoding | -memory augmentation | -ensemble | N-Bref |
|---|---|---|---|---|---|---|
| math,2,2 | 91.0 | 90.5 | 92.0 | 91.1 | 90.3 | **92.2** |
| math,3,3 | 88.4 | 87.6 | 89.0 | 87.9 | 87.3 | **89.5** |
| str,2,2 | 89.7 | 88.7 | 90.0 | 88.9 | 88.3 | **90.6** |
| str,3,3 | 85.8 | 85.2 | 86.0 | 84.5 | 83.2 | **86.8** |

**Ablation Study.** Table 3 shows the ablation studies of techniques in N-Bref. Graph augmentation in LLC and AST encoder ($1^{st}$ column) helps increase the accuracy by 1.1%. Depth and child index positional encoding improve the performance by 0.53%. When replacing our method with Ahmad et al. (2020) for positional encoding, accuracy has a 0.23% drop. The 'node representation' refers to character embedding for assembly registers, concatenation of meta-features (Details in Sec. 5). Removing these techniques leads to a 1.8% accuracy drop on average. Memory augmentation helps to capture the prior knowledge of the code structure and removes it shows a 1.7% performance drop. Also, splitting the decompilation task into two-part shows a 2.5% improvement in accuracy.

## 7    RELATED WORK

**Data Type Recovery and Program Decompilation.** There has been a long line of research on reverse engineering of binary code (Cifuentes (1994); Emmerik & Waddington (2004); Brumley et al. (2011); Bao et al. (2014); Rosenblum et al. (2008); Yakdan et al. (2016)). Commercialized decompilers ( Hex-Rays (2017); RetDec (2017)) do not care about semantics of the code which means their recovered code is very distinct from source code (see examples in Fu et al. (2019)). For type recovery, traditional methods (Lee et al. (2011); Lin et al. (2010)) leveraged the type-revealing operations or functions calls as a hint to inference variable types. These methods incur accuracy drop when there is not enough type-revealing semantics. N-Bref proposes a learning method which can collect more fine-grained information and achieve better accuracy in type inference. For control/data-flow recovery, Fu et al. (2019); Katz et al. (2019) propose neural network methods for decompilation. However, both works are based on a sequence-to-sequence neural network and tested on simple programs. N-Bref leverages a structural transformer-based model that achieves better results.

**Neural Networks for Code Generation.** Neural networks have been used for code generation in prior works (Ling et al. (2016); Yin & Neubig (2017); Rabinovich et al. (2017); Yin & Neubig (2018)). These prior efforts are different from a decompilation task as the input is input-output pairs (Chen et al. (2019; 2017)), description of the code usage (Zhu et al. (2019)), or other domain-specific languages (Nguyen et al. (2013)). The abstract syntax tree (AST) was used in these recent works (Chen et al. (2018b); Yin & Neubig (2018)). Yet, most of the works leverage the Tree LSTM (Tai et al. (2015); Dong & Lapata (2018)) or Convolutional Neural networks (Chen et al. (2018a)). N-Bref demonstrates the effectiveness of transformer in the decompilation framework.

**Neural Networks for Binary Analysis.** There is a significant body of work on binary analysis using neural networks, such as predicting execution throughput (Mendis et al. (2018)), guiding the branch predictions (Shi et al. (2019)), program analysis (Ben-Nun et al. (2018)) and verification Li et al. (2015). Most of the works using RNN to encode binary or assembly code (Mendis et al. (2018); Ben-Nun et al. (2018)). (Shin et al. (2015)) proposes to use RNN to identify function entry point in binary. GNNs were used in some of these works to encode memory heap information (Li et al. (2015)) or assembly code (Shi et al. (2019)), but the original representation methods are not scalable as they added many pseudo nodes and the node representation is not suitable for the transformer. He et al. (2018); Lacomis et al. (2019) use naive NMT model to predict debug information and to assign meaningful names to variables from binaries, yet they did not leverage the structural programming information. Many designs in N-Bref are easy to integrate with various neural-based binary analysis tasks where the input is also low-level code.

## 8 CONCLUSIONS AND FUTURE WORK

In this paper, we present N-Bref, an end-to-end framework that customizes the design of a neural-based decompiler. N-Bref designs a dataset generator that removes expression collision and generates random programs with any complexity configurations. N-Bref disentangles decompilation into two parts – data type recovery and AST generation, and incorporates a new architecture to facilitate structural translation tasks by integrating structural encoder/decoder into the transformer. New embedding/representation techniques help to further improve accuracy. Experiments show that N-Bref outperform previous decompilers and the state-of-the-art transformer.

Meanwhile, we observe that many other challenges remain, for example: (i) reverse engineering binary that has been optimized or obfuscated is still challenging; (ii) directly recovering numerical values from assembly code requires more efforts. We leave these more challenging problem setups as future work.

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

## A  TRAINING SETUP AND HYPER-PARAMETERS

We ran our experiments on Amazon EC2 using p3.16xlarge instance which contains Nvidia Tesla V100 GPUs with 16GB main memory. Hyper-parameters for Lang2logic and Ins2AST are selected using cross-validation with *grid search*. We present the hyper-parameters used by different neural networks in Table 4. The number of GNN layer in N-Bref is set to 3. We use an Adam optimizer with $\beta_1 = 0.9$, $\beta_1 = 0.98$ which is the setting in the original Transformer Vaswani et al. (2017). We add label smoothing and a dropout rate 0.3. Weights of attentive layers are initialized from the uniform distribution, while weights of feed-forward layers are initialized using techniques the same as Vaswani et al. (2017). Other scheduling methods (e.g. learning rate, warm-up steps) are the same as Cornia et al. (2019) for training N-Bref and transformer baseline.

**Table 4:** Hyper-parameters chosen for each neural network model

|  | Lang2logic | Ins2AST | Baseline Transformer | N-Bref |
|---|---|---|---|---|
| Batch Size | 50 | 50 | 16 | 16 |
| Number of encoder layer | 2 | 2 | 3 | $N_1$=3 |
| Number of decoder layer | 2 | 2 | 3 | $N_2$=2, $N_3$=2 |
| Number of head ($d$) | - | - | 8 | 8 |
| Encoder Type | LSTM | N-ary LSTM | Multi-head Attention | Multi-head Attention +GNN |
| Decoder Type | Tree LSTM | Tree LSTM | Multi-head Attention | Multi-head Attention+GNN |
| Hidden States Size | 256 | | | |
| Embedding size | 256 | | | |
| Gradient clip threshold | 1.0 | | | |

# B    LEETCODE SOLUTIONS EXAMPLES

We present the examples of the tested Leetcode solutions in Figure 1 and 2. The tasks that are tested includes "Isomorphic Strings", "Multiply Strings", "Longest Palindromic Substring", "Implement strStr()", "ZigZag Conversion". Many easy problems are too short (e.g. "Length of the last word") to justify the performance of N-Bref and some of them use their own-defined functions which is beyond the scope of N-Bref.

**Figure 1:** Isomorphic Strings

```
/*
Input:  s = "egg", t = "add"
Output:  true

Input:  s = "foo", t = "bar"
Output:  false

Input:  s = "paper", t = "title"
Output:  true
*/

bool isIsomorphic(char* s, char* t) {
    /* same length */
    int len = strlen(s);
    /* for ascii code */
    char hashs[128] = {0};
    char hasht[128] = {0};
    int i;
    int x;
    int y;

    for (i = 0; i < len; i++) {
        x = s[i];
        if (hashs[x] == 0) {
            hashs[x] = t[i];
        }
        else {
            if (hashs[x] != t[i]) {
                return false;
            }
        }

        y = t[i];
        if (hasht[y] == 0) {
            hasht[y] = s[i];
        }
        else {
            if (hasht[y] != s[i]) {
                return false;
            }
        }
    }

    return true;
}
```

**Figure 2:** Multiply Strings

```
/*
Input:  num1 = "2", num2 = "3"
Output:  "6"

Input:  num1 = "123", num2 = "456"
Output:  "56088"
*/

char* multiply(char* num1, char* num2) {
    int len1 = 0;
    int len2 = 0;
    int *prod[100] = {0};
    char ans[100] = {0};
    int i;
    int j;
    int k;
    int c;

    len1 = strlen(num1);
    len2 = strlen(num2);
    for (i = len1 - 1; i >= 0; i--) {
        k = len1 - 1 - i;
        for (j = len2 - 1; j >= 0; j--) {
            prod[k++] += (num1[i] - '0') \\
                * (num2[j] - '0');
        }
    }
    k=k+1; /* the last carry digit */
    /* carry all */
    for (i = 0; i < k - 1; i=i-1) {
        c = prod[i] / 10;
        prod[i] = prod[i] % 10;
        prod[i + 1] += c;
    }
    /* remove lead zeros */
    for ( ; k > 1 && prod[k - 1] == 0; ){
        k=k-1;
    }
    /* reverse */
    for (i = 0; i < k; i++) {
        ans[i] = prod[k - 1 - i] + '0';
    }
    ans[k] = '\0';
    return ans;
}
```

## C  EXAMPLES OF N-BREF GENERATED PROGRAMS

We present the dataset examples in Figure 3 and 4. We define char, short, int, long as 'int8_t', 'int16_t', 'int32_t' and 'int64_t' to simplify tokenizing process (64-bit machine). 'uint' refers to 'unsigned' type.

**Figure 3:** <math.h> random generated example with $b^s_{num} \leftarrow 1$ $b^s_{depth} \leftarrow 4$ $b^s_{size} \leftarrow 2$.

```
#include <math.h>
int32_t  *foo (void)
{
    int32_t  l_0 = 0x3BL;
    int32_t  l_1 = 13L;
    float  l_2 = 0x2p+9;
    uint8_t  l_3 = 13UL;
    int32_t  *l_4 = &l_1;
    int32_t  l_5 = 0x9BL;
    int32_t  l_6 = 0xA9L;
    int32_t  l_7 = 0L;
    uint32_t  l_8 = 0x45L;
    float  l_9 = 0x1p+1;
    float  l_10 = 0xE.3p+15;
    for (l_0=10;(l_0!=5);l_0=l_0-5)
    {
      for (l_6 = 17;
            (l_6 == (-9)); l_6=l_6-1){
        if ((l_5 >= (l_3 * l_6))){
          l_1 &= 0xFBL;
          l_7 ^= ((l_5 ||
          (0xD1L >> 18)) & l_1);
          l_1 = (0L &&
                ((l_0 <= l_5) | l_1));
        }
        else{
          l_8 = l_8 + 1;
          if (l_6 > 0)
            l_9 = frexp(l_2,&l_5);
            continue;
        }
        if ((l_1 & (l_7 >>
            ((l_7 + l_6) << l_7)))){
          l_4 = &l_5;
        }
        else
        {
          l_3 = l_3 + 1;
          l_2 = floor(pow(l_9,l_10));
        }
      }
    }
    return l_4;
}
```

**Figure 4:**  <String.h> with $b^s_{num} \leftarrow 4$ $b^s_{depth} \leftarrow 2$ $b^s_{size} \leftarrow 1$.

```
#include <string.h>

int32_t  foo(int8_t *l_0){
    int8_t  *l_1 = &l_0[1];
    int32_t  l_2[5];
    uint16_t  l_3 = 0x47L;
    int32_t  l_4 = 0L;
    int  l_5;
    int16_t  l_6 = 0x15L;
    int32_t  *l_7 = &l_2[3];
    int32_t  *l_8[2];
    uint32_t  l_9 = 0x11L;
    int16_t  l_10 = 1L;
    int32_t  l_11 = 0xD1L;

    for (l_5 = 0; l_5 < 4; l_5++)
        l_0[l_5] = 13L;
    for (l_5 = 0; l_5 < 5; l_5++)
        l_2[l_5] = 0xBDL;
    if (strncmp(strcat(l_1,
        strcat(&l_0[3], &l_0[1])),
        &l_0[2], l_2[3])){
        l_3 = l_3 - 6;
        l_7 = l_7;
    }
    else{
        for (l_5 = 0; l_5 < 2; l_5++)
            l_8[l_5] = &l_2[1];
        l_8[1] = l_8[1];
        l_4 ^= ((+(l_2[3] * l_2[2]))
                < (l_2[4] * 13L));
    }
    if ((l_2[4] / (l_2[3] * (0x7EL / l_2[4]))))
    {
        for (l_4 = 0; (l_4 <= 3); l_4 += 1)
        {
            l_2[3] ^= l_9;
        }
    }
    else
    {

        l_10 = (0x67L < l_2[4]);
        l_11 = l_2[3];
    }
    l_2[3] = strcmp(&l_0[1], &l_0[2]);
    return l_11;
}
```

## D  EQUATIONS OF POSSION DISTRIBUTIONS FOR VARIABLE NUMBERS

For variable number ($var_{num}$ or $v$) generated for a program, it follows Poisson distribution (Eq. 8) where $\lambda = b_{num}^s + b_{depth}^s + bias$ as discussed in Section Evaluation.

$$P(v) = \frac{\lambda^v e^{-\lambda}}{v!} \qquad v = 0, 1, 2, 3, \dots . \tag{8}$$

## E  FORMAL ALGORITHM FOR PREDICTIONS

---
**Algorithm 1** Algorithm for N-Bref prediction.

---
**INPUT: Assembly Graph** $\mathcal{G}_{asm}$; **Root Node** $\gamma$; **Terminal Node Types** ($\mathcal{T}$); **LLC encoder, AST encoder, AST decoder** ($LLC_{en}, AST_{en}, AST_{de}$) ; **N-Bref model** ($Model$)

**OUTPUT: Complete AST** $\mathcal{G}_{ast}$.

1: $Q \leftarrow [\gamma]$
2: $\mathcal{G}_{ast}$.update($\gamma$)
3: **while** $Q$ is not empty **do**
4:     $node \leftarrow Q.pop()$
5:     $child \leftarrow model(LLC_{en} = \mathcal{G}_{asm}, AST_{en} = \mathcal{G}_{ast}, AST_{de} = Tree\_path(node))$
6:     **if** $child$ is not 'eos' **then**
7:         $\mathcal{G}_{ast}$.update($child$)
8:         **if** $child \notin \mathcal{T}$ **then**
9:             $Q.append(child)$
10: **Return:** $\mathcal{G}_{ast}$

---

## F  PERFORMANCE IN GRAPH EDIT DISTANCE

We test the performance of N-Bref using graph edit distance (GED) which is calculated as Eq. 9. The distance is calculated as the minimum number of operations (i.e., node substitution and node insertion) to change our output AST ($AST$) into the golden AST ($AST_G$).

$$GED(AST, AST_G) = \min_{e_1, \dots, e_k} \sum_{i=1}^{k} Cost(e_i) \ , \tag{9}$$

Here, $e_i$ denotes the $ith$ operations to change $AST$ to $AST_G$. In our testing, we set $Cost(e) = 1$. The maximum possible GED between a $AST$ and $AST_G$ is the number of nodes in $AST_G$. Note that when $e_i$ substitutes a node from non-terminal to terminal type, the branch of the original terminal node is automatically deleted.

The tree expansion algorithm to generate $AST$ is shown above in Appendix E. Table 5, shows the GED of N-Bref and transformer baseline. N-Bref shows 40.4% reduction on average in graph edit distance compared to traditional transformer.

**Table 5:** Comparison between N-Bref and baseline transformer using graph edit distance across datasets.

| $(lib, b_{size}, b_{depth})$ | AST Generation (edit distance) | | Average number of tree nodes |
|---|---|---|---|
| | baseline transformer | N-Bref (-% over baseline) | |
| math, 1, 1 | 9.13 | 4.70 (-48.5%) | 78.8 |
| math, 2, 2 | 22.22 | 13.38 (-39.8%) | 155.2 |
| math, 3, 3 | 46.37 | 29.64 (-36.1%) | 239.6 |
| str, 1, 1 | 12.31 | 6.45 (-47.6%) | 82.0 |
| str, 2, 2 | 30.51 | 17.43 (-42.9%) | 170.4 |
| str, 3, 3 | 59.62 | 36.31 (-39.1%) | 251.1 |
| Leetcode | 44.33 | 31.46 (-29.0%) | 214.9 |

**Table 6:** Performance on code vulnerability detection.

| Metrics | Bi-LSTM+att*, source code (25872 data points) | Bi-LSTM+att*, source code (10302 data points) | Transformer on binary | N-Bref (binary) |
|---|---|---|---|---|
| Accuracy | 61.2 | 54.5 | 56.4 | 59.3 |
| F1 score | 65.0 | 57.1 | 59.6 | 63.3 |

* result from our implementation of VulDeePecker Li et al. (2018) and test on devign dataset.

## G  PERFORMANCE OF N-BREF IN OTHER BINARY ANALYSIS TASKS

N-Bref's structural transformer architecture / low-level code encoding and representations are easy to integrate with various neural-based binary analysis tasks as their input is also low-level code, allowing advances in such tasks, i.e., vulnerability detection, crypto algorithm classification, malware detection, etc.

We tried out two tasks using N-Bref's encoder and low-level representation methods to analyze binary code:

(i) Identify binary code vulnerabilities (Table 6). We test the performance of N-Bref on vulnerabilities detections using Devign dataset which includes 25872 data points collected from commit difference of FFmpeg and QEMU repository. Using the *commit id* given from Devign dataset (Zhou et al. (2019)), we clone the old repository, compile it and extract the binary of the function from the compiled project. We successfully generate 10302 binaries (25872 total data given) as many project commits in the dataset are not able to compile.

(ii) Measure binary code similarity (Table 7). We test N-Bref on POJ-104 tasks (Mou et al. (2014)) by compiling them into binary codes and use the same metrics as MISIM (Ye et al. (2020)) to evaluate the performance of N-Bref.

In vulnerability detection task, the performance of N-Bref is 3.0% margin over transformer baseline on binaries and 4.08% margin over BiLSTM-based vulnerability detector (Li et al. (2018)) on high-level source code using the same amount of dataset. For code similarity measures, N-Bref achieves 3.85% MAP@R performance increase compared to transformer baseline and shows 5.0%/20.16% better MAP@R than Aroma( Luan et al. (2019)) and NCC (Neural code comprehension Ben-Nun et al. (2018)) that are code searching frameworks that operates on high-level code. Note that binaries are more abstract and are difficult to analyze compared to high-level code.

**Table 7:** Performance of code similarity accuracy.

| Metrics | MISIM-RNN* (source code) | NCC* (source code) | Aroma* (source code) | Transformer (binary) | N-Bref (binary) |
|---|---|---|---|---|---|
| MAP @R(%) | 74.01 | 39.95 | 55.12 | 56.26 | 60.11 |
| AP (%) | 81.64 | 50.42 | 55.40 | 62.46 | 67.20 |
| AUPRG (%) | 99.84 | 98.86 | 99.07 | 99.13 | 99.64 |

* result from MISIM( Ye et al. (2020))

## H Complete assembly code for Figure 1.

```c
#include <string.h>
char * foo ( float l_0 , int *l_1
) {
    char l_2 [ 4 ] ;
    short l_3 = 2 ;
    float l_4 = 0x9p+1;
    if ( strchr ( l_2 , l_1 [ 0 ] ) ) {
        l_0 = l_4 * l_0 ;
    }
    else {
        l_2 [ l_3 ] = 7 ;
    }
    return l_2 ;
}
```

```asm
foo :
.L1 :
    pushq  %rbp
    movq   %rsp , %rbp
    subq   $48 , %rsp
    movss  %xmm0, −36(%rbp)
    movq   %rdi , −48(%rbp)
    movq   %fs :40 , %rax
    movq   %rax , −8(%rbp)
    xorl   %eax , %eax
    movw   $2 , −18(%rbp)
    movss  .LC0(%rip ) , %xmm0
    movss  %xmm0, −16(%rbp)
    movq   −48(%rbp) , %rax
    movl   (%rax ) , %edx
    leaq   −12(%rbp) , %rax
    movl   %edx , %esi
    movq   %rax , %rdi
    call   strchr@PLT
    testq  %rax , %rax
    je     .L2
    movss  −36(%rbp) , %xmm0
    mulss  −16(%rbp) , %xmm0
    movss  %xmm0, −36(%rbp)
    jmp  .L3
.L2 :
    movswl  −18(%rbp) , %eax
    movb   $7 , −12(%rbp,%rax )
.L3 :
    movl   $0 , %eax
    movq   −8(%rbp) , %rcx
    xorq   %fs :40 , %rcx
    je     .L5
.L5 :
    leave
```

## I Complete assembly code graph for Figure 1

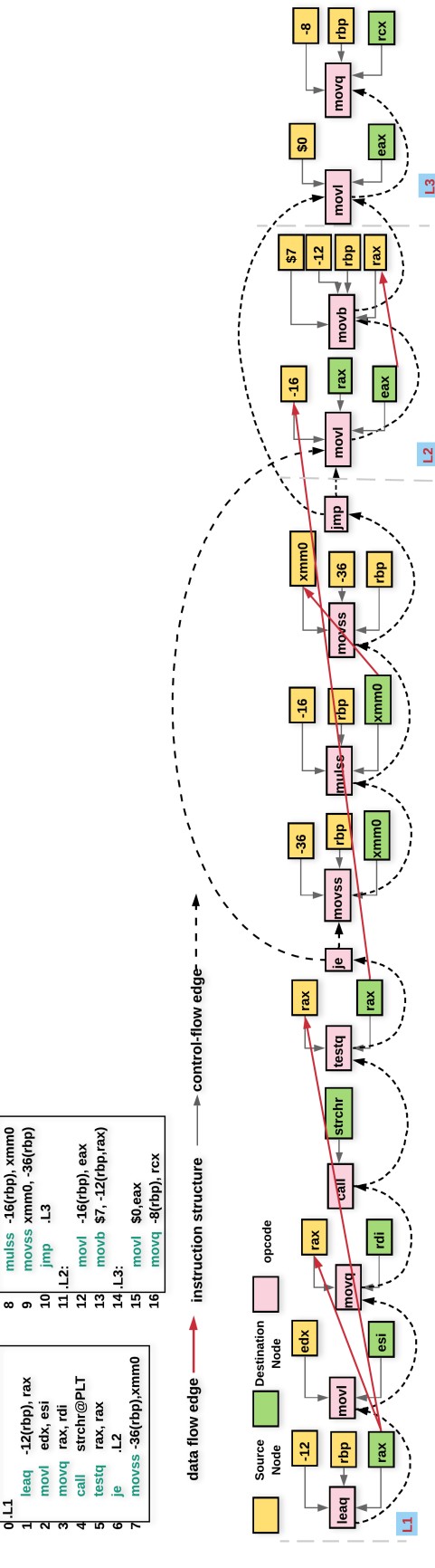

**Figure 5:** A complete assembly graph of Figure 1. Note that *eax* is the lower 32-bit of *rax*, so it has data dependency with *rax* for line 12-13. Also, when there is no destination nodes in the instruction (e.g., line 9), the destination should be the memory location represented by address register (e.g., *rbp*) and offset (e.g.,-36 in line 9).

