# OpenReview forum: "N-Bref : A High-fidelity Decompiler Exploiting Programming Structures "
_ICLR.cc/2021/Conference — Reject_

### Official Review · AnonReviewer1 · 2020-10-27
**architecture design**

**Rating:** 4
**Confidence:** 4

**Review:**


Summary:

The paper designs a neural architecture (called N-bref) for code decompiling, i.e., translating binary code to high-level code (e.g., C/C++).

Experiments show that a high token-level accuracy on some LeetCode dataset, compared with a few baseline methods including Seq2Seq and Transformer.


Major concerns:

1. The paper does not appear to be novel.

The paper identifies 4 challenges for decompiling, but some of them are generic, such as long-dependency problems and data augmentation. For the other two challenges (datatype and control/dataflow), the paper proposes to decompose the generation into two subtasks: source code generator (SC-Gen) and data type solver (DT-Solver).

Technically, both SC-Gen and DT-Solver are modeled by the same neural architecture (but differently parametrized): a memory-augmented structural transformer. This appears to be a ragbag of existing and known models: Transformer [Vaswani et al., 2017], Tree Transformer [Sun et al., 2020], and Memory Augmentation [Cornia et al., 2020]. I do not feel this paper very exciting.

2. The evaluation metric is problematic (or at least unclear).

The performance of a model is measured by "token accuracy" when the authors "expand the decompiled AST from the root into the same structure as the golden AST (AST_G)."

If this is what the authors meant, a real program is never generated. The authors assume a correct program is there, and predict the next AST token/rule assuming previous partial AST is the same as AST.

Such measure of success could be drastically different from the real performance of generation. The authors should consider string match of the generated program compared with reference, or the functional accuracy of the generated program.

With token accuracy given the correct partial AST, I would not agree with the claim that "N-Bref successfully reverts human-written" programs.

Minor:

For Ins2AST [Fu et al., 2019], did you use Ins2AST+Attention? That seems to be better than Ins2AST (w/o attention).

Decompilation tasks is --> Decompilation task is

---

> ### Author Response · Authors · 2020-11-17
> **Response to Review #1**
>
> We have updated our paper. Please refer to the thread “Revised Paper Uploaded” for a summary of major updates in our revision. Following is a detailed response to Review #1 comments.
>
> 1. **Novelty**:
>
> Please refer to the thread "Revised Paper Uploaded" above for our description of the novelty. We also add a summary of our contributions to the introduction of our revision. We emphasize our contributions of (1) new architecture and representation using domain-specific knowledge. (2) system integrating and tailoring existing NN components for special structural and long translation tasks. (3) thorough analysis and insight in encoder/decoder (4) configurable dataset.
>
> N-Bref also shows strong performance in other binary analysis tasks (Appendix G).
>
> 2. **Metric issue**:
>
> We show the comparison results in graph edit distance (for string matching as you suggested) without any constraints on prediction in Appendix F in revision. N-Bref shows a 40.4% reduction on average in graph edit distance compared to traditional transformers. The sequence generation manner of the traditional transformer fails to prevent error propagation at the early stage of the prediction.
>
> Two types of errors may occur in the program generation phase: structural errors (non-terminal-> terminal, terminal->non-terminal), and non-structural ones (terminal->terminal, non-terminal->non-terminal).
>
> Token accuracy directly reflects the precision of program structure recovery, since the predicted AST and the ground-truth one are represented as ordered and structural objects. As such, we believe token accuracy is a proper metric to measure the practical performance of binary decompilation.
>
> With the guidance of the token accuracy on AST generation, N-Bref achieves a high structural recovery precision. In particular, on average across the datasets, only 0.977% of the tokens show structural errors compared to the average error rate (8.92%) of N-Bref.

---

### Official Review · AnonReviewer2 · 2020-10-28
**The paper proposes a neural network based decompilation infrastructure for translation of assembly code to high-level source code. The paper seems promising, presents interesting ideas and promising results but is unfortunately hard to follow and fully understand.**

**Rating:** 5
**Confidence:** 3

**Review:**

The paper is interesting and addresses an interesting topic. The results seem promising, though the evaluation is done on relatively small test cases.

The proposal to divide the problem into two sub-tasks, i.e., data type solver and source code generation, is promising and can probably have impact on future decompiler proposals.

The main problem with the paper is that it is hard to read and understand. I'm aware of the page limitations, but the authors have crammed so much inte these pages that is hard to follow. Further, there are also a number of inconsistencies and unclear stuff (see below). For example, the authors claim that one thing their proposed method extends beyond earlier methods is to handle pointer references. However, I can't find in the paper how that is done.

The usefulness may be limited of this work, since they only work on unoptimized code (see e.g., page 4, 2nd paragraph). However, in reality, most code have went though substantial optimization during the compile phase.

Some other comments / questions:
* I lack information about the execution time of the training and inference.
* Page 2, 2nd paragraph. You claim that your code generator produces similar code styles as human programmers. What do you mean by that, and how du you support that claim?
* Fig 1b. Here are a number of strange / confusing things:
  - Why are not all asm instructions shown in the data flow graph? For example movl (line 2), call (line 4), testq (line 5), etc. are missing.
  - Why do you show one movss (line 9) and not the other movss (line 7)?
  - You have mixed up lines 7 and 8 in the graph (show it as mulss -36(rbp),xmm0), which is confusing.

---

> ### Author Response · Authors · 2020-11-17
> **Response to Review #2**
>
> We have updated our paper. Please refer to the thread “Revised Paper Uploaded” for a summary of major updates in our revision. Following is a detailed response to Review #3 comments.
>
> 1. **Training/evaluation time**:
>
> The training speed is ~41.6 samples/s (single V100) so training each epoch takes ~7 mins averaged across datasets. The inference is 91.9 samples/s. In practice, the training is parallelized across nodes using torch distribution data parallel interface.
>
> For binary analysis tasks using only the encoder (Appendix G), the training speed is 228.3 steps/s (single V100), and each epoch of training 22000 data points (size of POJ-104) takes ~1.4 mins.
>
> 2. **Explanation of ‘make code generated look like human-written code’**:
>
> We admit that there is no mathematical proof showing that our code is closer to a human-written one.
> Traditional code generators aim to debug compilers, so it involves a lot of complicated bit operations and does not have many diversities in the control flow. As such, we believe their codes are hard to interpret and do not match the diversity of human-written codes.
>
> **(i)** We add uniform distributions in sampling b_depth, b_size, b_num, and E_c (Poisson) as shown in Table 1 to cover more potential control flows of human-written codes.
>
> **(ii)** We adjust the uniform probability of generating each operator tokens, for example, “+, -,*” are more prevailing than “^, <<, >>,%”. We also adjust the probability of generating different standard library functions (some functions are more commonly used than others).
>
> 3. **Why do we disable compiler optimization?**
> **(1)** We disable the optimization for a fair comparison, as previous works [1,3,30] in the paper also disable optimizations.
>
> **(2)** Many optimizations change variable types and rewrite source codes. As such, the changed correct outputs will result in unfair accuracy evaluation.
>
> We believe that this is the first step to tackle this hard issue as the optimization destroys the semantics. We will leave the decompilation of more complicated configurations as future work.
>
> 4. **Explanation of Assembly graph (Figure 1)**:
>
> Regarding Figure 1 assembly graph, we update it and show the complete graph in Appendix I (finished in revision). We did not show all the nodes in Figure 1 due to page limits, and we showed only the instructions colored in green. In the training/evaluation, all the instruction nodes will be used.

---

### Official Review · AnonReviewer3 · 2020-10-28
**Nice results, somewhat chaotic presentation**

**Rating:** 7
**Confidence:** 3

**Review:**

The authors present a neural-based decompilation framework. They generate synthetic input data in order to make sure source code examples have a consistent code style that can be more easily learned. They predict types of variables and the actual code in two separate steps. For both steps, they employ a custom transformer architecture where evey second layer of the encoder is a graph neural network. There are two separate encoders of this kind, one conditions on a CFG obtained from static analysis of the input assembly, while the other one conditions on the partial output AST that has been generated thus far.

Strengths:
- The authors propose an end-to-end system for neural decompilation
- Interesting use of graph neural networks to increase sensitivity to structure in a transformer.
- Favourable comparison to multiple baselines.
- Evaluation also considers a human-written dataset.

Weaknesses:
- It is not so easy to fully understand the approach end-to-end. Perhaps the presentation can be improved. (Though I understand that it can be challenging to fit an explanation of an ambitious approach with multiple novel components into 8 pages.)
When reading the paper, it happened to me a couple times that I tried to go back to some piece of information and I did not find it at the location where I would expect to find it. Some details are discussed in the introduction, but apparently nowhere else, for example how the output of the DT-Solver is used. It would help to reorganize the paper a bit so that the exposition follows the order of operations when running the approach and to discuss which data goes where in the technical sections.
The paper says that DT-Solver and SC-Gen are both based on the same architecture, but DT-Solver is not really discussed in detail. As an example, it is not stated if the types of variables are chosen from some fixed set (which one?) or if the DT-Solver generates a type AST, but Figure 1 suggests a fixed set. Figure 3 is helpful, but it seems it does not show the full story for either DT-Solver or SC-Gen. Figure 1 is a bit confusing, as the shown AST does not appear to match the given source code. (E.g., there is no variable of type `int *`), and variable declarations are shown as part of a single AST of the program even though later they are treated separately.

- The evaluation metric is explained rather vaguely, so I am not sure if I fully understand what is meant, but this is crucial to interpret the results. How do you "expand the decompiled AST from the root into the same structure as the golden AST"? What happens during expansion if a token does not match? Is the subtree removed? I guess after the expansion step, you compare AST nodes that end up at the same position in the tree?


Further questions:

Accuracy based on syntax comparison to synthetically-generated input examples is not necessarily what an end user cares about. How well do your decompilation results preserve semantics? I.e., if token accuracy is imperfect, what kinds of mismatches do you typically get? It would also be interesting to understand a bit better the distribution of the results, e.g., how do the results change if you count the fraction of results with perfect token accuracy instead of computing averages over token accuracy?

As far as I understand, the positional encoding for ASTs drops a lot of structure information, which is then recovered by the GNN layers. Have you considered using richer positional encodings along the lines of [i]?

[i] https://www.microsoft.com/en-us/research/publication/novel-positional-encodings-to-enable-tree-based-transformers/


Minor:

Page 1: "learns to decompile the source code to assembly". That seems backwards.

Consider using \citep and \citet.

Please review your paper with a focus on grammar as well as whitespace and other formatting issues.
(For example, you should use ``$\mathit{xmm0}$`` instead of ``$xmm0$`` ,`` `control flow'`` instead of ``'control flow'``, etc.)

Page 14: There is wrong indentation or missing curly braces next to the "continue" statement in Figure 3.

---

> ### Author Response · Authors · 2020-11-17
> **Response to Review #3**
>
> We have updated our paper. Please refer to the thread “Revised Paper Uploaded” for a summary of major updates in our revision. Following is a detailed response to Review #2 comments.
>
> 1. **Clarification on the confusions**:
>
> We update Figure 1 and fix the error. In Figure 1, the second node should be ‘int_ptr’ (fixed in revision). . We add Sec 2.3 in the paper to explain the detail of DT-Solver and SC-Gen.
> DT-Solver only generates only the left part of the AST tree. The output of DT-Solver will be the initial tree input for SC-Gen. SC-Gen continues the expansion from the intermediate result of DT-solver. The DNN structure of DT-Solver is identical to SC-Gen.
>
> 2. **Metric**:
>
> We update the description of our evaluation metric and report the result of evaluation using edit distance without constraints on the structural expansion (see Appendix F). N-Bref shows a 40.4% reduction on average in graph edit distance compared to traditional transformers.
>
> For token accuracy, we expand the tree into the original structure. If the structural error occurs (e.g., non-terminal-> terminal nodes), we continue expanding the tree into the same structure. Note that structural error rarely happens. In particular, only 0.977% of the tokens show structural errors compared to the average error rate (8.92%) of N-Bref (on average). We thought accuracy is more interpretable than edit distance.
>
> 3. **Observations on error types**:
>
> Most of the errors are flipped tokens between non-terminal operators. For example, (1) ‘+’ and ‘-’ are confusing because ‘-’ can also be compiled into ‘add’, shr, ‘shl’ instructions depending on the variable type and if the variable is constant. (2) predicting the wrong variable id (e.g., var_1 -> var_0) is also a common error, because analyzing the numerical address offset to figuring out the variable id is difficult. We will leave a more thorough error analysis as future work.
>
> 4.  **Regarding novel positional encodings method mentioned**:
>
> We found the implementation of [i] as a function implemented here (https://github.com/microsoft/icecaps/blob/master/icecaps/util/trees.py). We tried out the encoding method on the AST encoder, which does not show performance improvement compared to our positional encoding method. (discussed in the Evaluation section). Note that this method still generates the output in a sequencing manner, which does not prevent error propagation like tree expansion.

---

### Official Review · AnonReviewer4 · 2020-10-30
**Review of N-Bref: a high-fidelity decompiler exploiting programming structures**

**Rating:** 3
**Confidence:** 4

**Review:**

High-level view:

I don’t think this is necessarily a bad paper, but I think it’s unacceptable for ICLR in its current form. I currently lean heavily toward rejection. After thinking over the concepts in the paper more, I might lean more strongly toward rejection or toward acceptance (if the authors can address the issues I raise below). I provide details below examination of how I’ve come to my evaluation rating below.

Summary:

This paper principally focuses on the idea of decompilation. Decompilation can mean many things, but the general idea as I understand it, is to take a representation of a software program from one level (e.g., program binary) and then “lift it” to a level that is higher in abstraction (e.g., from binary to assembly, from assembly to C, from C to a lambda calculus, etc.). As I understand it, it’s called decompilation because it tends to do the opposite of what a compiler does. Compilers tend to lower a representation of a software program into something that is closer to the hardware and therefore potentially more efficient.

The benefits of decompilation are numerous. One major benefit is in the ability to perform programming language – to – programming language transformation. Another, which is the focus of this paper, is for reverse engineering purposes of a binary. There are many others. As such, in my opinion this is unquestionably an important subtopic for the field of machine programming and the authors approach also seems satisfactory to me for ICLR (described below).

The authors present a new approach called: neural-based binary reverse engineering framework (N-Bref). N-Bref has a number of components that it relies on to perform its decompilation. They consist mostly of components from the programming languages community (e.g., assembly code, abstract syntax trees for encoding and decoding, etc.) and the machine learning community (e.g., deep neural networks for learning structural transformations, etc.).

The authors empirically evaluate their N-Bref’s accuracy on a number of problems from the open source LeetCode problem set and generate 25,000 pairs of high-level source and low-level source which are broken into training (60%), validation (20%), and testing (20%). LeetCode problems tend to be fairly simple, self-contained, and, to my knowledge, are coding problems that are meant to help train new programmers or prepare software developers for coding interviews, amongst other things. An emerging use of LeetCode is to use it as a baseline for machine programming (MP) in a variety of different ways. In this case, the authors are using LeetCode coded solutions in MP to compiled the source code into a lower level form (assembly I believe) and then see if N-Bref can return the assembly back to the original form or some semantically equivalent form. Their empirical approach seems sound to me.

Overall, the authors show better accuracy for their tested problem set against REWARD, a baseline system (a transformer), lang2logic, and Ins2AST across two dimensions: data type recovery and abstract syntax tree (AST) generation.

High-level concerns:

There are several reasons I’m not positive about this paper. Perhaps the biggest reason is I can’t seem to understand what is novel about the system. That is, unless I’ve just missed something, it seems that all of the core components of N-Bref are lifted from prior work with perhaps some minor augmentation. This feels largely incremental to me.

On the other hand, one could argue that N-Bref is novel because it combines a number of existing components in a unique way to achieve better performance that prior work. I can see this perspective. However, if we considered this view, it seems like the problem they are solving should be more impactful than type recovery and AST generation. I’m not saying these problems aren’t important – especially type recovery (I think this problem is deeply important) – but that it should go further to demonstrate more dimensions of decompilation.

The second major concern I have with this paper is the small dataset they are using. Consider, for a moment, that they are using only 25,000 input/output pairs for their training/validation/testing. Now consider a prior accepted ICLR 2020 paper, Hoppity (Dinella et al.), which trained on nearly 300k code change commits in GitHub. This looks like an order of magnitude difference in dataset empirical evaluation to me. On top of that, the only data is coming from LeetCode. We have no empirical demonstration that this approach will work on other datasets outside of LeetCode.

If the authors can address these two primary concerns by the time of decisions, I will likely slant toward the positive. If they do not (or will not), I will likely champion this paper’s rejection, as I do not believe in its current form it’s up to ICLR standards.

Low-level concerns:

The language in the paper seems to use many strong and ambiguous claims: “N-Bref outperforms previous neural-based decompilers by a large margin.” First of all, what is a “large margin”? There’s not quantitative measurement in the word “large”. Large could mean 1%, 10%, 100,000%. This kind of language is not what I expect from tier-1 publications.

Another example is: “However, key challenges remain:” where they then summarize two problems. I agree that the two problems they highlight are important. But I absolutely do not agree that those are the *only* two problems that stand in the way of decompilation.

Also, there seems to be some lack of understanding of the field of machine programming, from my perspective. For example in the abstract the authors claim “decompilation aims to reverse engineer binary executables.” I 100% disagree with this definition. As I stated above, I believe, the more general space of decompilation is actually the idea of lifting a software program representation from one format to a higher-level format that increases the level of abstraction from the hardware. Moreover, I know of many decompilation systems (e.g., verified lifting is one), that has an entirely different goal than reverse engineering. Verified lifting is principally focused, as I understand it, is focused on language to language translation.

Perhaps the grossest overclaim the authors make is in the introduction

“Our work pushes the performance of neural-based decompiler to a new level and presents a new baseline and stand dataset for future developers.”

I find that sentence simply unacceptable. I could never give an accept rating to a paper that makes such an outlandish claim with such a small body of evidence. Moreover, other people have used LeetCode as a baseline, so it’s not the first time people have done this. So it seems wrong to me on many levels.

This continues throughout the paper …

That said, these are minor nits that the authors, if they so choose, could probably fix with little effort.

I would hope that in a later version of the paper the authors would tone the language down, move away from the number of strong claims they make in the paper, and provide measurable data points when making claims about performance: “Our system is more accurate than <list the systems you’re comparing against> from X% to Y%.” Right now the only way to figure that out seems to be to deeply study the experimental evaluation, which is a bit inappropriate in my opinion. I believe it could (and should) be listed directly in the abstract and in the introduction. By hiding these details, it creates a perception of overclaiming – at least it did for me.

---

> ### Author Response · Authors · 2020-11-17
> **Response to Review #4**
>
> We have updated our paper. Please refer to the thread “Revised Paper Uploaded” for a summary of major updates in our revision. Following is a detailed response to R4 comments.
>
> 1. **Novelty**:
>
> Please refer to the thread “Revised Paper Uploaded” above for our description of the novelty. We also add a summary of our contributions to the introduction of our revision.
>
> 2. **Tone of Presentation**:
>
> We have improved the presentation and toned down the statements, as you suggested. We revise the description of our task to be ‘binary decompilation’. We omitted the quantitative results in the abstract of our original submission due to the page limit. The evaluation results are added in the abstract of our revised paper.
>
> 3. **Dataset size**:
>
> We thank the reviewer for pointing out the Hoppity paper. Hoppity’s dataset (~300K) focuses on fixing bugs; thus, each bug between commits is a point for training. For our translation tasks,  each AST node is a point for training. The average size of AST generated is around ~130 nodes (see Appendix F). Therefore, the total size of our training points is 25000*160 = 400K, which is comparable to Hoppity’s dataset and is sufficient for program translation.
>
> Moreover, our dataset size is similar to the ones used in other program translation/analysis tasks: for program translation employs 100K random generated data pairs but shorter code snippets compared to ours [i]; for vulnerability detections operates on ~12K data points for each dataset, which is sufficient to obtain high accuracy [ii].
>
> Furthermore, our token accuracy converges on 25K data points and shows sufficient generalization to other programs (see discussion of "Capability of generalization" below). We found that increasing the number of data points (50K and 70K) will increase accuracy by 0.91%/1.31% on average across datasets. However, doing so will significantly increase training time and computer memory consumption.
>
> Finally, our framework provides a data generator. Therefore, users can easily generate more data if needed. Data is very cheap in our case.
>
> 4. **Capability of generalization**
>
> Our dataset is randomized in multiple dimensionalities (Table 1). It can cover the majority of programs within the generation range.
>
> To justify the generalization capability of N-Bref, we further test the token accuracy of our model on the POJ-104 dataset [iii]. This is a dataset of human-written code that has a different distribution from our LeetCode test dataset. The N-Bref model trained on our math (2,2) dataset achieves 90.4% token accuracy on the unseen POJ-104 dataset.
>
> 5. **Impact of N-Bref on other binary analysis tasks**:
>
> Regarding the impact of the design, our structural transformer architecture, low-level code encoding, and representations are easy for integration with various neural-based binary analysis tasks whose input is also low-level code, allowing advances in such tasks (vulnerability detection, crypto algorithm classification, malware detection, etc.) Our decoder can be leveraged in other program synthesis tasks whose output is AST.
>
> We test N-Bref’s performance on two other binary analysis tasks in Appendix F:
>
> (Task 1) Identify binary code vulnerabilities.
> (Task 2) Measure binary code similarity.
>
> The performance of N-Bref on these tasks is also better than baseline transformers (by 3.0%/3.85% on Task 1 and Task 2, respectively), even beating some traditional methods that operate on high-level source code (by 4.1%/5% on Task 1 and Task 2, respectively).
>
> **References**:
>
> **[i]** Chen, Xinyun, Chang Liu, and Dawn Song. "Tree-to-tree neural networks for program translation." Advances in neural information processing systems. 2018.
>
> **[ii]** Zhou, Yaqin, et al. "Devign: Effective vulnerability identification by learning comprehensive program semantics via graph neural networks." Advances in Neural Information Processing Systems. 2019.
>
> **[iii]** Mou, Lili, et al. "Convolutional neural networks over tree structures for programming language processing." arXiv preprint arXiv:1409.5718 (2014).

---

### Author Response · Authors · 2020-11-17
**Revised Paper Uploaded**

We thank the reviewers for their valuable feedback. Major concerns include the representation, a need for detailed descriptions of our evaluation metric, unclear description of novelty, and small dataset size. We uploaded a revised version according to review comments. Following is a summary of our major updates (less major updates are described in response to each reviewer).

1. We tone down as suggested by reviewers in the abstract, introduction, and conclusions. We add the performance results back into the abstract and introduction.
2. We update Fig. 1. We also add a complete assembly graph in Appendix I. We clarify the usage of DT-Solver and SC-Gen in a new subsection (Sec.2.3, page 4).
3. We add two more new binary analysis tasks -- binary similarity check and binary vulnerability detection -- using the low-level code representation method and NN structure proposed by N-Bref in Appendix. G.
4. We update the description of our evaluation metric and present the result using a new metric (graph edit distance) in Appendix. F.
5. We fix other minor comments brought up by the reviewers.


**Novelty**: We make the following new contributions:

**(1) Models** A new structural transformer architecture with several new design principles that have not been explored in previous works: Existing transformer architectures are designed for conventional NLP tasks. The binary decompilation task that we target to solve involves long and highly structural input and output, which is more challenging than the NLP task.

To explore the intrinsic properties of low-level code input, we leverage domain-specific knowledge to optimize transformer design with memory and graph augmentation, disentangling data types recovery and AST generation. Moreover, we carefully design the representation of the low-level input and high-level code output as a graph and AST, respectively. Both the graph and the AST are suitable for N-Bref’s graph embedding and augmentation. Finally, we use hardware architecture knowledge, which has not been well explored in previous works, to develop a more efficient tokenizer for low-level code.

Furthermore, the above design optimizations can be easily adapted to other binary analysis tasks (Appendix F). We justify the effectiveness of N-Bref’s novel design compared to traditional transformers in Table 2, 5 (decompilation) and Table 6, 7 (new binary analysis tasks) in Appendix F.

**(2) System Integration**: We construct an end-to-end decompilation system by integrating the data generator / LLC Encoder / AST encoder and decoder in a holistic manner. Such integration has not been demonstrated in prior works. We bridge the gap between low-level and high-level code by transforming both into a graph space; these graph representations preserve syntactic and semantic information in the code.

**(3) Insights on encoder/decoder design**: We perform a comprehensive analysis of encoder/decoder design for decompilation tasks (Fig.4). Our analysis shows that the structural encoder/decoder has higher tolerance with increasing code length and complexity, compared to the naïve transformer or traditional methods.

**(4) New dataset**: Our configurable datasets uniform the code generation (removes Expression Collisions).

* The explanation of dataset size is in the response to Reviewer #4

---

### Decision · Program_Chairs · 2021-01-07
**Final Decision**

**Decision:**

Reject

**Comment:**

The paper describes N-Bref, a new tool for decompilation of stripped binaries. Compared to previous tools for neural-based decompilation, this tool is based on two new ideas: a) to separate the generation of data declarations from the generation of the code itself, and b) the use of more sophisticated network architectures. These network architectures, however, all come from prior work, so the contribution in that regard is only their application to this particular problem.

The authors addressed many of complaints raised by reviewers, particularly with regards to presentation and explanations, but I think the most substantial concerns remain.

The most substantial concern is novelty. The technique is built on a combination of existing models, and its only original idea seems to be to treat the generation of data declarations and the code itself as separate tasks to be handled by independently trained networks.

In terms of results, the paper shows some quantitative improvements over prior work, although it is not so clear that those improvements matter. The quality improvement is measured in terms of AST differences, but it is not clear how often those AST differences translate into semantic differences. More importantly, the tool is restricted to un-optimized binaries, which significantly limits its applicability for any real-world application. Prior work by Katz et al. is evaluated against optimized binaries, as are other types of lifting such as the Helium project by Mendis et al [1]. Given the prevasiveness of optimization in deployed code, a tool that cannot handle it has virtually no applicability.

I think some significant technical novelty could make up for the lack of evaluation against optimized binaries. Alternatively, strong results on optimized binaries would justify publication even if the technique is built from existing building blocks, but as it stands, I think the paper is too incremental to merit acceptance.

[1]  Charith Mendis, Jeffrey Bosboom, Kevin Wu, Shoaib Kamil, Jonathan Ragan-Kelley, Sylvain Paris, Qin Zhao, Saman P. Amarasinghe:
Helium: lifting high-performance stencil kernels from stripped x86 binaries to halide DSL code. PLDI 2015: 391-402